# Knowledge, Attitudes, and Perceptions of the Arabic-Speaking Community in Sydney, Australia, toward the Human Papillomavirus (HPV) Vaccination Program: A Qualitative Study

**DOI:** 10.3390/vaccines9090940

**Published:** 2021-08-24

**Authors:** Faeza Netfa, Catherine King, Cristyn Davies, Harunor Rashid, Mohamed Tashani, Robert Booy, S. Rachel Skinner

**Affiliations:** 1Discipline of Child and Adolescent Health, The University of Sydney Children’s Hospital Westmead Clinical School, Westmead, NSW 2145, Australia; catherine.king@health.nsw.gov.au (C.K.); cristyn.davies@sydney.edu.au (C.D.); harunor.rashid@health.nsw.gov.au (H.R.); mohamed.tashani@health.nsw.gov.au (M.T.); robert.booy@health.nsw.gov.au (R.B.); rachel.skinner@health.nsw.gov.au (S.R.S.); 2Kids Research, The Children’s Hospital at Westmead, Westmead, NSW 2145, Australia; 3National Centre for Immunisation Research and Surveillance, The Children’s Hospital at Westmead, Westmead, NSW 2145, Australia; 4Faculty of Medicine, University of Tripoli, Tripoli 13275, Libya

**Keywords:** adolescent, attitudes, cultural beliefs, knowledge, human papillomavirus, parents

## Abstract

Background: Little is known about acceptability of the human papillomavirus (HPV) vaccine among parents of adolescents from culturally and linguistically diverse backgrounds in Australia. This study aimed to explore the knowledge and attitudes of parents from Arabic backgrounds towards HPV vaccination offered to their children in the national school-based vaccination program. Methods: Qualitative interviews were conducted in Western Sydney, with parents of adolescents from Arabic backgrounds. Recruitment was via informal personal contacts and passive snowballing. Face-to-face semi-structured interviews were conducted in Arabic. These were audio-recorded, transcribed, and translated into English. Thematic analysis was used to identify emerging themes. Results: Commonly identified themes across fifteen interviews included: (1) lack of awareness and knowledge of HPV and its vaccination, (2) awareness and understanding of the government vaccination information sheet, (3) parents’ preferences for information provision, (4) the role of parents’ religious beliefs in forming attitudes about HPV vaccination, and (5) lost opportunities to educate parents about HPV vaccination during general practitioner (GP) visits. Conclusion: The findings point to the need to address cultural, language, and communication barriers to improve awareness and acceptability of HPV vaccination in the Arabic community. Educational strategies should be tailored to this community based on their specific information needs and preferences.

## 1. Introduction

Coverage of recommended vaccinations is lower in new immigrants to Australia [1,2]. Little is known about the acceptability of the human papillomavirus (HPV) vaccine among parents of adolescents from culturally and linguistically diverse (CALD) backgrounds [1,2]. It is therefore essential for public health prevention programs to target new immigrants who may have not been vaccinated against HPV, using culturally sensitive approaches.

HPV infection is a sexually transmitted disease that affects both males and females, which can spread through genital contact [3,4]. The low-risk types of HPV cause benign genital warts, whereas the high-risk types can cause cancer of the cervix, vulva, vagina, penis, anus, and throat [3,5,6,7,8]. Worldwide, cervical cancer is the fourth most common cancer affecting women, the seventh most common cancer overall, and the fourth leading cause of cancer death in women, with approximately 570,000 cases and 311,000 deaths in 2018 [5,9,10,11].

Vaccination against HPV together with HPV screening in adult women is the most effective way to reduce cervical cancer incidence and mortality [12,13,14]. HPV vaccination can prevent persistent infections and significantly reduce the incidence of HPV-associated precancerous lesions [13,15]. In 2007, Australia was the first country to introduce a national publicly funded immunisation program with a quadrivalent HPV vaccine for girls aged 12 and 13 as part of National Immunisation Program (NIP) administered in schools [16]. From 2013, boys were also included in the program [17]. Since 2018, the HPV vaccine changed from the quadrivalent to nonavalent vaccine administered using a two-dose schedule. The National School-Based HPV Vaccination Program has achieved generally high uptake [18]. However, there is variability in uptake across schools in Australia.

Knowledge and understanding of HPV infection and HPV vaccination are important factors affecting decision-making about the vaccine [19,20,21]. In 2020, our systematic review involving 13 studies revealed that parent immigrants and refugees in western countries had limited knowledge about HPV infection and vaccination. We identified 13 studies: four studies reported initial negative attitudes, such as fearing that HPV vaccination would encourage sexual activity. However, we also found that once parents received vaccine information, their negative attitudes often changed [22]. Poor knowledge and understanding of HPV and the HPV vaccine is also well-documented in the broader Australian community and most other populations [23]. Low knowledge may be a contributor to poor HPV vaccine acceptance [24].

In Australia, people from racial and ethnic minorities may be referred to as culturally and linguistically diverse (CALD), which includes anybody born overseas in countries other than those classified as primarily English-speaking. Australia has had increased language, cultural, and religious diversity following successive waves of immigration [25,26]. Census data revealed that there were over 7.5 million migrants living in Australia in 2019, with 29.7% of the population born overseas [25]. The 2016 Census also showed that there were over 300 separately identified languages spoken in Australian homes and 21% of Australians spoke a language other than English at home. Arabic was reported as the third most common language spoken at home (after English and Mandarin) [27].

This study aimed to explore the HPV vaccine knowledge, attitudes, and practices of parents from Arabic Muslim backgrounds in Western Sydney, Australia, towards HPV vaccination offered at school to their adolescents aged 12–14 years. We aimed to investigate parental reasons for acceptance and non-acceptance of the HPV vaccine and to understand the perceived barriers and facilitators to HPV vaccination.

## 2. Materials and Methods

Participants, setting, and recruitment: Semi-structured interviews were conducted face-to-face with participants by the first author who is formally trained in qualitative methodologies (FN). The interviews took place in Western Sydney, Australia (which is one of the most CALD areas of New South Wales (NSW) [28], approximately 50 km from the Central Business District of Sydney). Participants were mothers from Arabic Muslim backgrounds, whose adolescents (12–14 years of age) were offered the HPV vaccine in schools. There were two key recruitment strategies. First, parents of HPV-vaccine-eligible adolescents, including friends and acquaintances of the first author (FN), from the Arab community were invited to participate in the study. Second, additional parents were invited to participate via passive snowballing. Participants who completed an interview were then asked to circulate the research information among their friends. When interested parents contacted the researcher, they were provided with additional study information via telephone. Participants received the participant information sheet (PIS) and the consent form via mail or email.

Data collection: We used purposive sampling to select 15 participants. Interviews were conducted exclusively in Arabic by the first author (FN), who is bilingual, from October 2018 to July 2019. Since the principles of ethnography include building trust [29,30], the interviewer took steps to ensure participants felt comfortable discussing personal and health issues. This included organising the interviews at a place and time convenient for the participants and communicating in the participant’s preferred dialect. The interview schedule included questions about knowledge, attitudes, and beliefs concerning the HPV vaccine. Interviews were audio-recorded, transcribed verbatim, and then translated into English. Transcripts were reviewed by two bilingual experts to assure consistency of translation.

Data analysis: Data analysis took place concurrently along with data collection to allow researchers to refine the interview questions with subsequent participants. Qualitative data from the transcripts and translations were coded and categorised according to inductive thematic analysis using NVivo 12 software [31,32] Codes were reviewed by three co-authors (SRS, CK, and CD) to ensure consistency of coding. Thematic analysis was used to identify emerging themes and sub-themes. Data were compared to identify recurring themes and issues. Themes were documented, including de-identified, illustrative comments from participants. Conceptual saturation was reached when no new themes or subthemes were generated from the data.

## 3. Results

### 3.1. Participants’ Characteristics

Parents: Fifteen mothers, all of whom were Muslim, participated in interviews. The majority of interviews were conducted one on one, apart from one interview with two mothers who wished to be interviewed together. The countries of origin for participants were as follows: Algeria (*n* = 3), Djibouti (*n* = 1), Lebanon (*n* = 1), Libya (*n* = 4), Iraq (*n* = 3), and Palestine (*n* = 2) (Table 1). One participant was born in Australia but had spent most of her life in Lebanon. The median duration of the interviews was 32 min (range 24–54 min). As most participants had difficulty speaking and understanding English, all interviews were undertaken in Arabic. Demographic information is provided in Table 1.

Themes: We identified five key themes highlighting the knowledge, attitudes, and perceptions of parents from an Arabic background in Australia towards the HPV vaccine offered to their adolescents: (1) lack of awareness and knowledge of HPV and HPV vaccination; (2) awareness and understanding of the government vaccination information sheet; (3) parents’ preferences for information provision; (4) the role of parents’ religious beliefs in forming attitudes about HPV vaccination; (5) lost opportunities to educate parents about HPV vaccination during general practitioner (GP) visits.

#### 3.1.1. Lack of Awareness and Knowledge of HPV and HPV Vaccination

We found that Australian parents of Arabic background had poor understanding about HPV and HPV vaccination. Most parents had not heard of the HPV virus or HPV-related diseases prior to the interviews, and many felt that they did not have adequate information about HPV:

*“No, I didn’t hear that cancer can be caused by a virus. I heard that cancer could be hereditary, but not caused by a virus. I didn’t know that viruses can cause cancer”*.KAP008

Although one parent knew that HPV is a virus that causes warts, she did not know it can cause genital warts or sexually transmitted infections (STIs):

*“My prior knowledge is that it causes warts, but I didn’t know that it causes sexually transmitted infections and different types of cancer, this is new information that I have just read in the information sheet you have provided to participate in this research”*.KAP001

Mothers reported that they did not have detailed knowledge of HPV and did not know that it is a sexually transmitted infection. Mothers did not know that the HPV vaccine protects adolescents from HPV-related cancer. Some participants had heard that the HPV vaccine can protect females from cervical cancer. However, none of the parents had any knowledge or information about the potential for the HPV to cause cancer in males. Mothers commented:

*“I have no information that it affects men, all I know is that it protects the partner, which stops the transfer of the virus. Boys are vaccinated to protect the female partner by preventing the virus from being transferred”*.KAP001

*“Yes, I know that this cancer affects women, but I did not know that it can be caused by a virus, and how can this virus affect men?”*.KAP008

Most parents were unaware that the HPV vaccine was recommended specifically for adolescents. Most participants accepted the HPV vaccine for their adolescent as it was considered a childhood vaccine:

*“I didn’t expect that one of these vaccines was an HPV vaccine to prevent the STI. I thought that it was childhood vaccines, which means that I didn’t know that she was vaccinated with a different vaccine from her childhood”*.KAP0010

Most parents reported that they did not have sufficient information about the HPV vaccine to make a decision about HPV vaccination for their adolescent. Many of these parents reported seeking out other information sources including relatives or friends, media, and Internet websites to acquire information. Mothers who had basic information about the HPV vaccine indicated that it prevents cervical cancer and it may protect their daughters in the future when they are married.

#### 3.1.2. Awareness and Understanding of the Government Vaccination Information Sheet

There were significant barriers hindering parental access to accurate evidenced-based information about HPV vaccination such as not receiving the government information sheet from their adolescent’s school. The main source for mothers to gain knowledge about the HPV vaccine was the information sheet, written in English, which was sent from the schools via students. Some parents reported that they did not receive the information sheet from schools, as some students forgot to give this to their parents. Two mothers commented:

*“The school didn’t send the information sheet. I don’t know, I didn’t know if the school send the information sheet or no”*.KAP002

*“The school sent the information sheet and Consent Form, but my daughter forgot to give it to me to sign on it”*.KAP004

Another barrier that parents identified was that the information sheet was in English. Many participants in this study revealed that their limited proficiency in the English language affected their confidence in making a decision for their adolescent to have the HPV vaccine. Several participants referred to language as a main barrier as they had difficulty in understanding technical terminology. One mother commented:

*“I understand English but when I read an information for example on vaccination, I prefer it in Arabic language. Possible if I read the topic in English, I can’t understand it 100%; sure, if I read it in Arabic, it would be clearer for me”*.KAP005

One participant reported her experience with a translated information sheet from English to Arabic, which she said was difficult to understand and inaccurate. She observed:

*“The note about the vaccine was in English, which I prefer because Arabic translation sometimes is inaccurate, and I have few experiences that translated Arabic is unintelligible, so I prefer English where I understand more as you always have a feeling that there is something wrong with the translation”*.KAP003

Participants also reported becoming aware of the HPV vaccine from Centrelink letters. Centrelink is an Australian Commonwealth Government agency that provides financial support and other social services to the community. Centrelink sends letters to parents when a child or adolescent is missing any vaccine doses. Centrelink may suspend family benefits if parents refuse to give permission for their adolescent to have the HPV vaccine [33].

*“I have received a Centrelink letter stating that my son has some outstanding vaccines that he has not taken, which resulted in suspension of Centrelink payment. If he didn’t take them Centrelink payment would have been stopped altogether. When they sent the letter, I knew that my son didn’t take the vaccine”*.KAP002

#### 3.1.3. Parents’ Preferences for Information Provision

Participants reported that they needed more information about HPV and the HPV vaccine in Arabic language. Participants also needed detailed information about the benefits, side effects, and ingredients of the HPV vaccine. Most participants prefer an SMS message from school to remind them about vaccination day for their adolescent. Others wanted to have the Information Sheet sent via mail. Due to the language barrier, most mothers suggested more information to be obtained from experts in a face-to-face meeting at school; this could be conducted in Arabic or with a skilled interpreter.

*“I wish by Arabic language or in Arabic and English language, because there are medical terms that are hard to understand it in English language, but in Arabic language I can read it and sign on it. Many times, my daughter brings letters from school, it is written in English language and I don’t know what is in it”*.KAP0011

*“I would like that an expert explained it at the school. It could be through the parents meeting, via face-to-face meeting, and I would like this session in the Arabic language”*.KAP0012

## 4. The Role of Parents’ Religious Beliefs in Forming Attitudes about HPV Vaccination

Our study found that religious beliefs played an integral role in these parents’ attitudes towards HPV vaccination uptake for their adolescents and were at times a barrier against approving HPV vaccination. One mother reported that she rejected all vaccines for her children as she believes that God created man in the best way and shape. In her view, anything that contradicted this belief suggests that we have a problem in our bodies.

*“In our religion, we believe that God created human in the best manner and shape, so anything interferes with this shape, like vaccine, which is injected into our body, it is kind of belief that we have problem in our body”*.KAP0014

### 4.1. Parents’ Attitude to Halal Vaccine

Islamic law prohibits the use of medicines or ingredients from non-Halal (non-permissible) sources. Many parents expressed concern that HPV vaccines may be contaminated with protein from pigs, which would prohibit their use for Muslim families. Some participants accepted the HPV vaccine as Halal was mentioned on the top of information sheet, which encouraged them to accept the HPV vaccine for their adolescents. Many mothers indicated that they would not give permission to their adolescents to have the HPV vaccine if it was a non-Halal vaccine. One mother reported:

*“I saw Halal at the top of the information sheet, the vaccine components are Halal, and most of the school staff members were Muslims of Arab background, the word Halal was written”*.KAP001

### 4.2. Parents’ Religious Beliefs Concerning Sexual Relations

The importance of religion in terms of accepting or rejecting the HPV vaccine was raised by participants. Some parents described their shock upon learning that the HPV vaccine protects against a STI.

*“We were shocked when we heard that the HPV vaccine is to protect from STI, as it is curious that vaccine protects against STI at this age”*.KAP0012

For some, this was perceived as a conflict to their religious beliefs. Participants had doubts about the HPV vaccine as they were concerned it may keep their children away from their religion commandments; therefore, some participants declined the HPV vaccine for their children. Some mothers commented that in their religion sex outside marriage is a sin, therefore, they declined the vaccine.

### 4.3. Parents’ Attitudes Differed by Gender of Adolescent

Our study found that some mothers responded differently towards the HPV vaccine depending on their adolescent’s gender. Some participants preferred to give the HPV vaccine to their son but not to their daughters. One mother reported:

*“I would have vaccinated my son, but I would not give it to my daughter though as son is different to daughter, I cannot guarantee that my son will not do anything because he has the freedom to come and go, a daughter stays under the auspices of her parents, she does not come and go alone, and she knows that this thing is not part of our religion”*.KAP002

We also found that most mothers were able to discuss HPV vaccination and protection from diseases with their daughters but described difficulties and embarrassment in discussing that with their sons. This was especially when talking about the protection the vaccine provides against STIs. One mother commented:

*“My son is different to my daughter, when he brought vaccination paper, I had a quick look at it, he didn’t say anything to me, neither did he tell me about information given to him at school. I did not ask him; my son gets shy with me and I didn’t give him any information either”*.KAP001

## 5. Lost Opportunities to Educate Parents about HPV Vaccination during GP Visits

### 5.1. Inadequate Communication with GPs

Most participants with poor English skills wanted to understand the content of the information sheet. When going to their GPs to gain more information about the HPV vaccine, they reported that their GP did not inform them that HPV vaccine can protect against STIs. Most mothers reported that their GP just recommended it without providing adequate explanations. One mother described that there is a lack of proper communication between Australian doctors and immigrants.

### 5.2. Parental Views about GP, Health Provider Communication

The topic of sex was sensitive, so GPs spoke with mothers without raising that HPV is sexually transmitted. Many mothers indicated that they wanted to know more about the vaccine and to understand which diseases against which the vaccine offers protection. One mother reported:

*“GP didn’t tell me anything. He did not explain about these vaccines (year 7 vaccines), maybe he assumed that I know about the vaccines and I thought he knows better about the topic, he should have explained to me”*.KAP002

## 6. Discussion

To our knowledge, this is the first study to identify the knowledge, attitudes, and perceptions of parents from Arabic backgrounds in Australia toward the HPV vaccine for their adolescents. We found a lack of knowledge and understanding about HPV and HPV vaccination among Arabic mothers. Generally, knowledge of Arabic mothers about HPV disease and the HPV vaccine was limited, and most participants had insufficient knowledge to answer questions the researcher posed about HPV and the HPV vaccine. Our study indicates that although the HPV vaccine has been in use for over a decade, information about this vaccine did not appear to have been well-disseminated to these mothers.

Many mothers were not sure whether their children received the HPV vaccine, and some had no idea whether they gave approval for consent. Some mothers expressed limited knowledge about other HPV-related cancers, with only one mother mentioning that she has knowledge about HPV in relation to cervical cancer. This study has identified a need for community education about HPV-related cancers, which will help to assist in clarifying the relevance of the HPV vaccine. Previous studies conducted in Australia that measured student knowledge of HPV and risks associated with cervical cancer found that students’ knowledge about HPV and cervical cancer was incomplete [23,34]. In this study, some mothers reported that they have difficulties in talking to their sons about how the HPV vaccine can protect from a STI. These findings highlight the need for community education about HPV-related disease and the HPV vaccine in a culturally safe and accessible approach. This education could possibly be provided in school information sessions as requested by parents. If HPV vaccination for adolescents was more widely understood and discussed in these communities, this may assist in reducing the discomfiture that mothers may encounter while discussing the vaccine with their sons.

Additionally, our study showed that language was a barrier that hindered mothers’ understanding of the HPV vaccine. It was difficult for mothers to access accurate information about the HPV vaccine presented in an accessible way. These factors have also been identified as important in previous studies in western countries [23,35,36,37,38,39]. Our research found that parents perceived inaccuracies in the Arabic translation of the information sheet that was sent to parents from schools. The Arabic language exists in two forms, a spoken form, which consists of home country dialects, and standard Arabic, which consists of both a spoken and a written form. The participants reported that the information sheet was mostly written in one of the several home country dialects; this dialect is cumbersome for some parents to understand. Parents in this study reported a strong preference to receive information about the HPV vaccine verbally from experts who speak Arabic language or with a translator. Of note, vaccination providers’ recommendations have been found to be a significant facilitator of HPV vaccination in previous studies in the USA and Australia [40,41,42]. It is likely that the recommendation of an Arabic-speaking doctor to explain vaccination would be important for parents of Arabic-speaking backgrounds.

Our study demonstrates that parents are keen to hear about HPV and the HPV vaccine at school through a parent meeting or during GP visits, preferably in Arabic language. Further, information addressing religious concerns such as the Halal status of the vaccine must be made a priority in communications with this community. This has been well-illustrated by other studies conducted in Malaysia (2009, 2010, 2018) [43,44,45]. Awareness of the HPV vaccine among mothers in this study and their perception of its role in prevention of HPV-related disease was low. A study in Saudi Arabia also reported that that awareness and perceptions of young women towards the HPV vaccine was relatively low [46].

A key strength of this study was that the interviewer spoke Arabic language in different regional dialects, which made it possible to recognise slight differences in expression that took place during the interviews. This also allowed for a clearer discussion during the interview with participants. We reached data saturation with 15 participants, with no new themes or subthemes emerging from the data.

Sharing the same cultural background as the participants might have caused some important information to be omitted, as the participants used a shorthand way to refer to some issues. However, the researcher addressed this issue by asking the participants to provide more detail and extra information about their opinions when their answers seemed to be obvious. A weakness of this study was that only mothers were recruited, as in Arabic Muslim culture it is inappropriate for a female interviewer to speak to fathers about culturally sensitive issues. Future studies should include a male interviewer to capture data from fathers with HPV-vaccine-eligible adolescents to seek their views and experiences. Additionally, as this initial study was conducted in one area of Sydney only, we suggest expanding future efforts to Arabic-speaking participants in other geographic areas to gain a greater understanding of the needs of this population.

## 7. Conclusions

The research findings point to the need to address language, cultural, and communication barriers in order to improve the awareness and acceptability of the HPV vaccine among the Arabic community in Australia. Access to the Arabic version of the HPV information sheet needs to be facilitated in a meaningful translation that allows parents to understand what is written about HPV disease and the HPV vaccine. Educational strategies should be tailored to this community based on their specific information needs through parent meetings and face-to-face seminars with experts who can speak Arabic.

## Figures and Tables

**Table 1 vaccines-09-00940-t001:** Demographics of study participants.

Participant Number	Age in Years	Ethnic Background	Education Level	Year of Migration to AUSTRALIA	Child’s Sex	HPV Vaccine Received	Information Sheet Received from School
KAP001	42	Arabic-Libyan	PhD	2007	Girls	Yes	Yes
KAP002	46	Arabic-Lebanese	Year 7	2006	Girl & Boy	Yes	No
KAP003	37	Arabic-Palestinian	University	2014	Boy	Yes	Yes
KAP004	45	Arabic-Iraqi	High school	2010	Girl	Yes	No
KAP005	48	Arabic-Libyan	University	2001	Girl	Yes	Yes
KAP006	45	Arabic-Djibouti	Year 8	1995	Boy	Yes	Yes
KAP007	42	Arabic-Algerian	University	1998	Boy	Yes	Yes
KAP008	46	Arabic-Palestinian	University	2014	Boy	Yes	Yes
KAP009	34	Arabic-Lebanese	Year 12	Born in Australia	Boy	Yes	Yes
KAP0010	42	Arabic-Iraqi	University	2013	Girl	Yes	Yes
KAP0011	40	Arabic-Libyan	Intermediate Institute (equivalent of Year 12)	2010	Girl &Boy	Yes	Yes
KAP0012	38	Arabic-Libyan	1st year University	2006	Girl	Yes	Yes
KAP0013	44	Arabic-Iraqi	University	1998	Boy	Yes	Yes
KAP0014	40	Arabic-Algerian	PhD	1996	Girl	No	Yes
KAP0015	40	Arabic-Algerian	University	2001	Boy	No	Yes

## Data Availability

The data presented in this study are available on request from the corresponding author. The data are not publicly available due to ethical restrictions.

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
