# Peer review of "Knowledge, Attitudes, and Perceptions of the Arabic-Speaking Community in Sydney, Australia, toward the Human Papillomavirus (HPV) Vaccination Program: A Qualitative Study"

_vaccines, 2021, doi:10.3390/vaccines9090940_

Round 1
Reviewer 1 Report
The Paper takes a very interesting topic, I agree with the authors that effective prophylaxis against STIs needs acceptance of parents to HPV vaccination of adolescents. The acceptance of vaccination and awareness is related to the knowledge and understanding of the infectious meaning for future cancer developments.
In my opinion, a serious weakness of the presented paper is the small group of studied parents and used type of sampling. The authors explain that they decided to use the snow bowling method and I don't understand why? To recognize parental opinion will be better to organize a cross-sectional type of study with a higher number of participants. Could the authors explain what is the participation ratio? How many invited parents to agree to the interview? On the other hand, I would like to know what is the frequency of HPV vaccination in Muslims and other nations in Australia?
All of the subjects are women, most of them have university graduation (only 3 have a low level of education). No men in the studied respondents. I have a question for the authors, who of parents decide about childrens’ health (e.g. vaccination) in Muslim families, mother or father? Maybe it will be an interesting point of view in the discussion.
Author Response
REVIEWER 1
Comments and Suggestions for Authors
The Paper takes a very interesting topic, I agree with the authors that effective prophylaxis against STIs needs acceptance of parents to HPV vaccination of adolescents. The acceptance of vaccination and awareness is related to the knowledge and understanding of the infectious meaning for future cancer developments.
In my opinion, a serious weakness of the presented paper is the small group of studied parents and used type of sampling. The authors explain that they decided to use the snow bowling method and I don’t understand why? To recognize parental opinion will be better to organize a cross-sectional type of study with a higher number of participants. Could the authors explain what is the participation ratio? How many invited parents to agree to the interview? On the other hand, I would like to know what is the frequency of HPV vaccination in Muslims and other nations in Australia?
All of the subjects are women, most of them have university graduation (only 3 have a low level of education). No men in the studied respondents. I have a question for the authors, who of parents decide about children’s health (e.g vaccination) in Muslim families, mother or father? Maybe it will be an interesting point of view in the discussion.
Query 1: In my opinion, a serious weakness of the presented paper is the small group of studied parents and used type of sampling.
Our response:
Our paper uses qualitative, not quantitative research methods. In contrast to quantitative research, qualitative research does not aim to produce generalisable findings at a population level. Instead, qualitative methods are used to explore an issue in-depth with a certain population group. Sample size calculations are not a feature of qualitative research, rather, data are collected until data saturation is reached on a topic, with no new themes being identified. Qualitative research often uses small numbers of interview subjects. Research suggests that the more homogenous a group, the smaller the number of interviews required to reach data saturation. Methodological studies have suggested data saturation is reached in such groups with as few as 6-12 interviews (Guest, Bunce, Johnson. How Many Interviews Are Enough?: An Experiment with Data Saturation and Variability. Field Methods. https://journals.sagepub.com/doi/10.1177/1525822X05279903)
Our research uses a very homogenous and well-defined population of interest – immigrant mothers speaking Arabic, who have children aged 14-16 in one geographic area (Western Sydney, NSW, Australia).
We used a combination of snowballing and purposive sampling, both of which are well-accepted methods in qualitative research.
Initially recruited participants were asked to circulate information brochures about the research project among their friends. Interested participants then contacted the researcher. This was a non-confrontational and culturally sensitive way to locate participants for this study. Purposive sampling was then used to ensure the researcher included both mothers that were accepting of HPV vaccination and those that were not.
This type of purposive sampling is commonly used in qualitative studies to provide richly textured information and to collect a manageable amount of data.
We reached data saturation with 15 participants, with no new themes or subthemes emerging from the data.
Query 2: The authors explain that they decided to use the snowbowling [sic.] method and I don’t understand why?
Our response:
Snowballing is a well-recognised method used in qualitative research to recruit participants. As this study was on a topic of potential sensitivity – HPV – a sexually transmitted infection, care was taken to provide a non-confrontational recruitment method. Snowballing allowed participants to provide information about the study to their broader contacts, which is much more acceptable in this community than hearing about the study from someone they have never met. The study initially asked recruited participants who had completed an interview to provide research information, to their friends. Interested parents then contacted the researcher.
Query 3: To recognize parental opinion will be better to organize a cross-sectional type of study with a higher number of participants.
Our response: As this research is the first study to examine the knowledge, attitudes and perceptions of Arabic speaking mothers in Australia toward the HPV vaccine we chose qualitative methods to try to carefully elicit any concerns or knowledge gaps in this population. We were not able to commence with a cross-sectional study, as initial research was required to explore issues of concern this community may have about the HPV vaccine. Conducting exploratory qualitative research before designing a cross-sectional survey reflects best practice. Without such an understanding, any questions we designed may not have been reflective of community concerns. Also, in contrast to quantitative research, qualitative research allows refinement of question schedules to pursue emerging themes in more depth with subsequent participants.
The findings from this study can be used to inform future studies using other methods, such as cross-sectional studies.
Query 4: Could the authors explain what is the participation ratio?
Our response: We do not know the participation ratio as this is not a usual feature of qualitative recruitment. As we mentioned in our response to Q1, the initially recruited participants who had completed an interview were asked to circulate research information brochures among their friends. Interested parents contacted the researcher. It is not known how many research information brochures were distributed among parent networks.
Query 5: How many invited parents to agree to the interview
Our response: Parents who were interested and eligible as volunteers to participate in this research contacted the researcher.
Participants were enrolled by one of the following two strategies:
1) From informal personal contacts: This group included friends and acquaintances from the Arabic community who usually meet at public places in Western Sydney. The researcher, Faeza Netfa put the flyers on a table located at the meeting place and asked the parents if they wish to collect the flyers and read them later. If any of those parents were interested and eligible as volunteers to the research study, they then contacted the researcher to be interviewed.
2) Passive snowballing: The initially recruited participants were asked to circulate the flyers and to provide research information to their friends. Interested parents contacted the researcher.
Query 6: I would like to know what the frequency of HPV vaccination in Muslims and other nations in Australia is
Our response: Vaccination status is not routinely collected using religious background; thus, it is unknown how many Muslims in either Australia or the world have been vaccinated against HPV. Many predominantly Islamic countries do not yet have a fully established national HPV vaccination program (Global HPV Vaccine Introduction Overview May 2020. https://path.azureedge.net/media/documents/Global_HPV_Vaccine_Intro_Overview_Slides_webversion_2020May.pdf)
Since the HPV vaccine was licensed, there has been an increase in immigrants from different cultures and languages coming to Western countries. Most of the immigrants originate from socio-economically low to intermediate level countries, and do not have a nationally funded HPV vaccination program, therefore, it is reasonable to believe that most immigrants do not have background knowledge about HPV vaccination.
Coverage of recommended vaccinations is lower in new immigrants to Australia (References 1- 2 in the manuscript).
Query 7: No men in the studied respondents.
Our response: We added sentences to the limitation paragraph: Line 339- 340
We recruited only women for our study as our study focuses on a sensitive topic for this community, given that the HPV vaccine protects against a sexually transmitted infection. The cultural norms in this community around sensitive topics prevent a female researcher (FN) asking fathers about this sensitive topic (about HPV and transmission of HPV, the protection of children from STIs and sexual relationships).
We agree with the reviewer that in the future, it would be good to have male participants in a study on this topic, with interviews conducted by an Arabic-speaking male researcher.
We have added text to the limitations section of the paper to reflect this issue:
[One limitation of this study was that as per cultural norms, only mothers could be recruited to be interviewed by a female researcher. It was less appropriate for a female researcher to ask questions to fathers about a sexually transmitted infection (HPV).
In the future it will be good to explore these issues with fathers in the Arabic-speaking community, with interviews conducted by an Arabic-speaking male researcher.]
Query 8: I have a question for the authors, who of parents decide about children’s’ health (e.g. Vaccination) in Muslim families, mother or father? Maybe it will be an interesting point of view in the discussion.
Our response: We found that in our study it was mostly Arabic mothers who signed the consent form to allow their children to have HPV vaccine. Some mothers relied on their husbands to sign the consent form if they did not understand English.
Query 9: On the other hand, I would like to know what is the frequency of HPV vaccination in Muslims and other nations in Australia?
We have addressed this issue in Q6 above.
Reviewer 2 Report
Dear authors
Analysis by paper partitions:
1 - Introduction: the content and the drafting of the general part must be reformed to review the syntax of the topic
PMID: 32344551 ; PMID: 29943384; PMID: 31193477
2- Discussion: to deepen the use of essential oils as a new frontier in Papilloma virus therapy in view of the problem of HPV resistance. Find out more about this aspect (max two lines) using and citing the following references:
3 - Check the bibliographic entries throughout the text, some of which do not conform, review some entries in the references and necessarily insert those referred to in point 2 for the purpose of acceptance by me.
4 - Review English grammar and in particular applied scientific English: in particular, tenses and syntax in the discussion
Author Response
Reviewer 2:
Query 1 - Introduction: the content and the drafting of the general part must be reformed to review the syntax of the topic
Our response: We made some modifications to the text to improve the language and grammar. This has been done by several co-authors who are native English speakers.
PMID: 32344551; PMID: 29943384; PMID: 31193477 as references for the essential oil.
Query 2- Discussion: To deepen the use of essential oils as a new frontier in Papilloma virus therapy in view of the problem of HPV resistance. Find out more about this aspect (max two lines) using and citing the following references.
Our response: This study is a qualitative study to describe knowledge, attitudes and perception about HPV and vaccination in an Arabic ethnic group in Australia. It does not aim to evaluate or assess any treatments for HPV and such an assessment is beyond the scope of our study. We thus respectfully submit that it would not be appropriate for us to try to discuss the treatment of HPV virus using essential oil therapies in this manuscript. Citing references about essential oil therapies for HPV disease is beyond the scope of our research and this manuscript.
Query 3 - Check the bibliographic entries throughout the text, some of which do not conform, review some entries in the references and necessarily insert those referred to in point 2 for the purpose of acceptance by me.
Our response: Thank you for this feedback- we have reviewed the references used throughout the text. In regards to the 3 references about the use of essential oils as a treatment for HPV – as mentioned above, this is beyond the scope of our study, which is a qualitative study on HPV and acceptance of HPV vaccine in the Arabic-speaking mothers in Sydney, Australia.
Query 4 - Review English grammar and in particular applied scientific English: in particular, tenses and syntax in the discussion
Our response: As mentioned above, we have made some modifications to the text to improve the language and grammar. This has been done by several co-authors who are native English speakers.

Round 2
Reviewer 1 Report
Many thank authors for the comments and explanation, now I better understand the intention of the researchers. However, I propose some modifications:
- I expect, that the authors can change a title of the paper adding at the end that these are preliminary results of a qualitative research
- In the section of the Material and methods authors should be cited the article (Guest, Bunce.. ) to supporting the choosen number of subjects (e.g in the line 98)
Author Response
Thank you for the feedback have provided on our manuscript.
Response(s) to reviewers’ comments
REVIEWER 1
Comments and Suggestions for Authors
Many thank authors for the comments and explanation, now I better understand the intention of the researchers. However, I propose some modifications:
1. I expect that the authors can change a title of the paper adding at the end that these are preliminary results of a qualitative research
2. In the section of the Material and methods authors should be cited the article (Guest, Bunce.. ) to supporting the chosen number of subjects (e.g in the line 98).
Query 1: I expect that the authors can change a title of the paper adding at the end that these are preliminary results of a qualitative research
Our response 1:
We thank the reviewer for this comment. We have adapted the title to be more precise about our interview population and location of participants and added that it is a qualitative study.
The modified title is as follows: Knowledge, attitudes and perceptions of the Arabic-speaking community in Sydney, Australia toward the Human Papillomavirus (HPV) vaccination program: a qualitative study
We have also suggested in the discussion that as this is an initial study only, that further research should be conducted on this topic with participants from a broader geographic area: [Also, as this initial study was conducted in one area of Sydney only, we suggest expanding future efforts to Arabic-speaking participants in other geographic areas to gain a greater understanding of the needs of this population.]
Query 2: In the section of the Material and methods authors should be cited the article (Guest, Bunce… ) to supporting the chosen number of subjects (e.g in the line 98)
Our response 2:
Thank you for this suggestion. We have added 2 references as per the below:
1. One as per the reviewer’s request: Reference 29. [Guest, G., et al. How many interviews are enough? An experiment with data saturation and variability. Field Methods 2006, 18, 59-82] in line 58
2. Another reference: Reference 21. Davies, C., et al. School-based HPV vaccination positively impacts parents’ attitudes toward adolescent vaccination. Vaccine 2021, 39:4190-4198] in line 99
Query 3: Review English grammar and in particular applied scientific English:
• English language and style: Reviewer 1 comment:
• (x) Extensive editing of English language and style required.
Our response 3:
This manuscript has been extensively reviewed by co-authors who are native English speakers. Tracked changes were provided in the initial response to reviewer’s manuscript, outlining where text had been changed to improve clarity. We have also improved the title to improve English language usage and add greater clarity about the study.
Query 4: Does the introduction provide sufficient background and include all relevant references? Can be improved
Our response 4:
Thank you for this comment. We have now added an additional reference to the Introduction:
Reference 21. [Davies, C., et al. School-based HPV vaccination positively impacts parents’ attitudes toward adolescent vaccination. Vaccine 2021, 39:4190-4198] in line 99.
Query 5: Are the methods adequately described? Reviewer 1 comment: Can be improved
Our response 5:
We added a range of details to the Methods in the initial response to reviewers including:
• Addition of a statement about purposive sampling
• Addition of details about when the interviews were conducted
• Addition of details about conceptual saturation of data.
We have also now added a further reference to additionally enhance the Methods section - [Guest, G., et al. How many interviews are enough? An experiment with data saturation and variability. Field Methods 2006, 18, 59-82].